# Photodynamic Inactivation of Bacteria and Biofilms with Benzoselenadiazole-Doped Metal-Organic Frameworks

**DOI:** 10.3390/molecules27248908

**Published:** 2022-12-14

**Authors:** Liang Luan, Lehan Du, Wenjun Shi, Yunhui Li, Quan Zhang

**Affiliations:** 1Department of Laboratory Medical Center, General Hospital of Northern Theater Command, No. 83, Wenhua Road, Shenhe District, Shenyang 110016, China; 2Institute of Smart Biomedical Materials, School of Materials Science and Engineering, Zhejiang Sci-Tech University, Hangzhou 310018, China

**Keywords:** antimicrobial agents, bacteria, biofilms, irradiation, metal-organic frameworks

## Abstract

Bacterial biofilms are difficult to treat due to their resistance to traditional antibiotics. Although photodynamic therapy (PDT) has made significant progress in biomedical applications, most photosensitizers have poor water solubility and can thus aggregate in hydrophilic environments, leading to the quenching of photosensitizing activity in PDT. Herein, a benzoselenadiazole-containing ligand was designed and synthesized to construct the zirconium (IV)-based benzoselenadiazole-doped metal-organic framework (Se-MOF). Characterizations revealed that Se-MOF is a type of UiO-68 topological framework with regular crystallinity and high porosity. Compared to the MOF without benzoselenadiazole, Se-MOF exhibited a higher ^1^O_2_ generation efficacy and could effectively kill *Staphylococcus aureus* bacteria under visible-light irradiation. Importantly, in vitro biofilm experiments confirmed that Se-MOF could efficiently inhibit the formation of bacteria biofilms upon visible-light exposure. This study provides a promising strategy for developing MOF-based PDT agents, facilitating their transformation into clinical photodynamic antibacterial applications.

## 1. Introduction

In recent decades, bacterial resistance to antibiotics has occurred frequently due to the misuse and overuse of antibiotics, leading to a failure to treat bacterial infectious diseases [1,2,3]. In addition, most bacterial infections are related to the formation of biofilms. Generally, bacteria can secrete extracellular polymeric substances such as lipids, proteins, and polysaccharides and embed themselves to form a cohesive network containing multiple cells [4,5]. Compared to separated bacteria, bacterial biofilms are 10–1000 times more resistant to antibiotics [6,7]. Although numerous antibacterial agents and antiseptic techniques have been proposed, there remains a pressing need for new antibacterial reagents that are not dependent on antibiotics and can increase the efficiencies of killing pathogens resistant to antibiotics and inhibiting the formation of bacterial biofilms [8,9,10].

Photodynamic therapy (PDT) has been widely studied as a promising treatment strategy for bacteria ablation because it does not cause bacterial resistance to light treatment [11,12,13]. The PDT process begins when a photosensitizer (PS) is activated by light of the appropriate wavelength to transfer energy to molecular oxygen and produces highly cytotoxic reactive oxygen species, primarily singlet oxygen (^1^O_2_), which damages the membrane and DNA of bacterial cells and leads to cell death. To date, various organic PSs have been developed, including porphyrins [14,15], phthalocyanines [16,17], and boron dipyrromethene [18,19]. However, the use of organic PSs is often limited by their aggregation in physiological environments, leading to the loss of photosensitizing activity [20,21,22,23]. Inorganic nanomaterials can also act as PSs to kill bacteria under light irradiation but exhibit severe toxic effects on specific mammalian cells [24,25,26].

Metal-organic frameworks (MOFs) are a class of porous materials composed of inorganic metals and coordinating organic ligands [27,28,29]. Recently, MOFs have been widely explored for biomedical applications due to their unique properties, including tunable composition and size, large surface area, and high porosity for the delivery of various therapeutic agents [30,31,32]. When PSs are incorporated into MOF skeletons, various photosensitive MOFs have been developed for photodynamic antibacterial applications. For example, porphyrin-based MOFs have been explored in bacterial treatment, as they can avoid the self-quenching of PSs and facilitate the diffusion of ^1^O_2_, thus improving the PDT efficacy of photosensitive MOFs [33,34,35]. Although various photosensitive MOFs have been studied for photodynamic antibacterial applications, there is still a need to develop a simple strategy for preparing photosensitive MOFs, which can effectively inhibit the formation of bacterial biofilms.

In this study, we report a strategy for preparing zirconium (IV)-based benzoselenadiazole-doped MOFs with mixed ligands and demonstrate their effectiveness for the photodynamic inactivation of bacteria and biofilms. The structure and preparation strategy of the MOFs are illustrated in Figure 1. The dimethyl-substituted dicarboxylate ligand (mTPDC-H2) was first synthesized to construct a topological UiO framework (Me-MOF). Furthermore, the benzoselenadiazole-containing ligand (SeTPDC-H2) was synthesized and mixed with mTPDC-H2 to form the benzoselenadiazole-doped MOF (Se-MOF). To endow MOFs with photodynamic properties, the introduction of selenium atoms in the Se-MOF can effectively exert the heavy-atom effect that enhances the intersystem crossing of the exciting energy. In addition, the incorporation of benzoselenadiazole-containing ligands into MOF skeletons can also avoid the aggregation of organic photosensitizers [36]. The photodynamic antibacterial activity of Me-MOF and Se-MOF against *Staphylococcus aureus* (*S. aureus*) bacteria was studied. Finally, the bacterial biofilm model was established to evaluate the inhibition effect of Me-MOF and Se-MOF on biofilm formation under visible-light irradiation.

## 2. Results and Discussion

The synthetic processes of mTPDC-H2 and SeTPDC-H2 are illustrated in the Appendix A, and their chemical structure was determined by ^1^H NMR spectroscopy (Appendix A). Using acetic acid (HAc) as an additive, the mixture solution of ZrCl_4_ and mTPDC-H2 in N,N’-dimethylformamide (DMF) was heated at 105 °C for 48 h. The resulting precipitation was collected by centrifugation and washed three times with DMF to give Me-MOF. The morphology of Me-MOF was observed by scanning electron microscopy (SEM). As shown in Figure 2A, Me-MOF has an octahedral morphological structure, and its average diameter is approximately 1.2 µm. Furthermore, mTPDC-H2 and SeTPDC-H2 were mixed in a mole ratio of 4:1 to prepare the Se-MOF using similar methods. It can be seen in Figure 2B that there was no difference in morphology between Me-MOF and Se-MOF. Furthermore, the porous structure on the surface of the MOF was observed in the enlarged SEM images of both samples.

The formation of MOF structure was confirmed using Fourier transform infrared (FT-IR) spectroscopy. The FT-IR spectra of Me-MOF and Se-MOF were measured and compared with those of mTPDC-H2 and SeTPDC-H2 (Figure 2C). For mTPDC-H2 and SeTPDC-H2, there was a characteristic peak at 1690 cm^−1^ due to the stretching vibrations of the carboxyl C=O group. However, the intensity of the peak at 1690 cm^−1^ decreases significantly and two new peaks at 1540 and 1417 cm^−1^ were observed in the FT-IR spectra of Me-MOF and Se-MOF, indicating the formation of a coordination bond between the Zr^4+^ and carboxyl groups [37]. Powder X-ray diffraction (XRD) patterns of Me-MOF and Se-MOF are shown in Figure 2D. Three characteristic peaks at 2θ = 4.54°, 5.24°, and 9.16° were observed for two MOF samples, and their relative intensity was consistent with the simulated pattern from single-crystal data, confirming their regular crystallinity and UiO-68 topological framework [38,39].

The nitrogen adsorption/desorption isotherms were measured to verify the porous feature of the MOFs. As shown in Figure 3A, a typical type I reversible isotherm was observed for the Me-MOF and Se-MOF. According to Brunauer-Emmett-Teller (BET) analysis, Me-MOF and Se-MOF have surface areas of 2991 and 2882 m^2^ g^−1^, respectively. As shown in Figure 3B, an average pore diameter of ~2 nm was calculated by Barret-Joyner-Halenda (BJH) analysis. Next, two MOF samples were dispersed in DMF, and fluorescence spectra were measured at room temperature. Compared to Me-MOF, Se-MOF has a longer excitation wavelength in the range of 400–475 nm (Figure 3C). Furthermore, the maximum fluorescence emission of Se-MOF was at 523 nm, but no fluorescence was observed for Me-MOF (Figure 3D).

UV-vis absorption spectra of mTPDC-H2 and SeTPDC-H2 in DMF were also recorded (Figure 3E). The results show that SeTPDC-H2 has both UV and visible light absorption (260–470 nm), while mTPDC-H2 only has UV absorption (260–330 nm). Thus, a blue light-emitting diode (LED) lamp (450 nm) was used as the light source to stimulate the photoactivity of Se-MOF in the following experiments. Next, electron spin resonance (ESR) was used to evaluate the efficiency of ^1^O_2_ generation for Me-MOF and Se-MOF under 450 nm light irradiation with a density of 3 mW cm^−2^ for 5 min. First, two MOF samples were dispersed in phosphate-buffered saline (PBS) solutions (pH 7.4) at a concentration of 1.0 mg mL^−1^, and 2,2,6,6-tetramethylpiperidine (TEMP) was used to monitor ^1^O_2_ generation before and after irradiation. As shown in Figure 3F, no noticeable change in the ESR signal was observed for Me-MOF after irradiation, which should be attributed to the fact that mTPDC-H2 has no absorption at 450 nm. However, a much stronger characteristic signal of ^1^O_2_ was observed in the ESR spectra of Se-MOF under light irradiation. These results show that incorporating SeTPDC-H2 into the MOF skeletons enables Se-MOF to generate ^1^O_2_ under 450 nm light irradiation, indicating the application potential of Se-MOF as a photodynamic agent.

The photodynamic antibacterial activity of Me-MOF and Se-MOF against *S. aureus* was evaluated using a two-fold serial dilution method [40,41,42]. The bacteria were incubated with different concentrations of MOF materials, followed by irradiation with a LED light (450 nm, 3 mW cm^−2^) for 20 min. After light irradiation was repeated three times, the bacterial suspension was diluted and cultured on Luria-Bertani (LB) agar plates. As shown in the digital photographs (Figure 4A), the number of bacteria was calculated according to ImageJ software and analyzed for comparison. As shown in Figure 4B, the results showed that the photoactivity of Se-MOF was significantly higher than that of Me-MOF under identical conditions. This outcome should be because Se-MOF produced ^1^O_2_ under 450 nm light irradiation, thus causing more significant photodamage to bacterial cells. In addition, the dark toxicity of Me-MOF and Se-MOF against *S. aureus* was also studied. In the absence of light, no differences in bacterial viability were observed between Me-MOF and Se-MOF (Figure 4C). Thus, these results further demonstrated the potential of Se-MOF for antibacterial PDT applications.

The bacterial biofilm model was established to investigate the inhibition effect of Se-MOF on biofilm formation under visible light irradiation. Both PBS and Me-MOF were used as control samples. Since Me-MOF has the same physical and chemical properties as Se-MOF, Me-MOF was selected as the control to avoid interference by other factors (controlled release of antibacterial components, sizes, morphologies, surface charges, and interaction with the bacterial cell wall) on bactericidal performance [43,44]. First, *S. aureus* bacteria were incubated in the medium containing Me-MOF or Se-MOF at an equivalent concentration of 50 µg mL^−1^. After being irradiated three times with a LED light (450 nm, 3 mW cm^−2^) for 20 min, the bacterial suspension was cultured on 96-well plates. The biofilm inhibition efficiency of Se-MOF at different time points was studied by crystal violet (CV) staining. Representative images of the plate are presented in Figure 5A and the optical density (OD) at 590 nm was measured using a microplate reader. The data indicated that Me-MOF has no inhibition effect on the biofilms of bacteria under 450 nm light irradiation. However, Se-MOF significantly inhibited the formation of bacteria biofilms under identical conditions. As the ^1^O_2_ produced by Se-MOF inhibited bacterial reproduction, it destroyed bacterial aggregation to form biofilms. Therefore, these results confirmed that Se-MOF could be used as photosensitive MOFs for the photodynamic killing of bacteria and inhibition of biofilm formation.

## 3. Materials and Methods

### 3.1. Synthesis of mTPDC-H2 and SeTPDC-H2

The synthesis procedures are described in the Appendix A.

### 3.2. Preparation of Me-MOF

mTPDC-H2 (200 mg, 0.58 mmol) was dissolved in DMF (100 mL), and then ZrCl_4_ (150 mg, 0.64 mmol) and acetic acid (3 mL, 0.05 mmol) were added to the solution. The mixture was heated to 105 °C and stirred for 48 h. Then, the crude product was collected by centrifugation and washed three times with DMF (100 mL) and ethanol (100 mL), respectively. Finally, the sample was dried under reduced pressure and denoted as Me-MOF.

### 3.3. Preparation of Se-MOF

mTPDC-H2 (140 mg, 0.4 mmol) and SeTPDC-H2 (44 mg, 0.1 mmol) were dissolved in DMF (100 mL). After adding ZrCl_4_ (120 mg, 0.51 mmol) and acetic acid (3 mL, 0.05 mmol) to the solution, the mixture was heated to 105 °C and stirred for 48 h. Then, the crude product was collected by centrifugation and washed three times with DMF (100 mL) and ethanol (100 mL), respectively. Finally, the sample was dried under reduced pressure and denoted as Se-MOF.

### 3.4. Antibacterial Assay

Monoclonal colonies of *S. aureus* grown in the LB agar plate were transferred to an LB culture medium (10 mL) and grown at 37 °C for 12 h. After the bacterial suspension was diluted to 10^5^ CFU/mL (OD_600nm_ = 0.001), 100 µL of the diluted bacterial suspension was incubated with different concentrations of Me-MOF or Se-MOF in a 96-well plate. The plate was irradiated with a blue LED light (450 nm, 3 mW cm^−2^) for 20 min, and the irradiation was repeated three times. After the bacteria were further incubated at 37 °C for 12 h, the bacterial solution was diluted 10^−5^ times and 100 µL of the diluted bacterial solution was streaked on LB agar plates using the spread plate method. After the plates were cultured for 12 h, the number of colony-forming units was counted by ImageJ software.

### 3.5. CV Staining of Bacteria Biofilms

The bacterial suspension was diluted to 10^7^ CFU/mL (OD_600nm_ = 0.1), and 100 µL of the diluted bacterial suspension was incubated with Me-MOF or Se-MOF at an equivalent concentration of 50 µg mL^−1^ in a 96-well plate. The plate was irradiated with a blue LED light (450 nm, 3 mW cm^−2^) for 20 min, and the irradiation was repeated three times. At different times of incubation (6, 12, and 24 h), the biofilms were observed and quantified using a CV staining method [45,46,47]. The biofilms were fixed with 4.0% paraformaldehyde and then stained with an aqueous CV solution (0.1% *w*/*v*) for 30 min. After removal of the CV solution, the stained biofilms were washed three times with PBS (pH 7.4) and then dissolved in an ethanol solution (95% *v*/*v*). Finally, the optical density at 590 nm (OD_590nm_) was measured using a microplate reader (Varioskan LUX, ThermoFisher SCIENTIFIC, Waltham, MA, USA) to determine biofilm biomass.

## 4. Conclusions

In summary, we have synthesized Se-MOF as an antibacterial material for the photodynamic inactivation of bacteria and biofilms. The obtained Se-MOF is a topological UiO-68 framework with a diameter of ~1.2 µm. The results demonstrated that incorporating benzoselenadiazole-containing ligands into MOF skeletons endowed them with high efficiency to produce ^1^O_2_ under 450 nm light irradiation, causing photodynamic inactivation of *S. aureus* bacteria. In particular, Se-MOF could efficiently inhibit the formation of bacterial biofilms upon visible-light irradiation. Thus, Se-MOF may be promising MOF-based PDT agents for the photodynamic killing of bacteria and inhibition of biofilm formation.

## Figures and Tables

**Figure 1 molecules-27-08908-f001:**
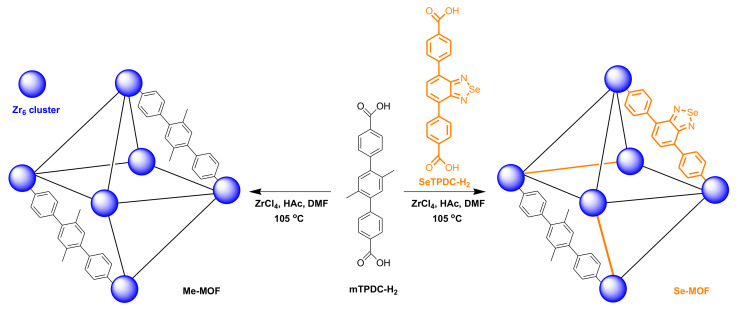
Schematic representation of the preparation for Me-MOF and Se-MOF.

**Figure 2 molecules-27-08908-f002:**
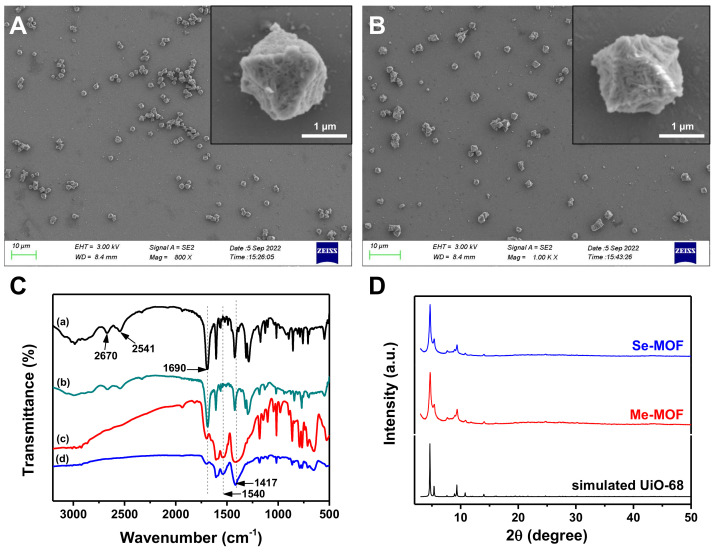
SEM images of (**A**) Me-MOF and (**B**) Se-MOF. (**C**) FT−IR spectra of (a) mTCDP-H2, (b) SeTPDC-H2, (c) Me-MOF, and (d) Se-MOF. (**D**) XRD patterns of the Me-MOF and Se-MOF.

**Figure 3 molecules-27-08908-f003:**
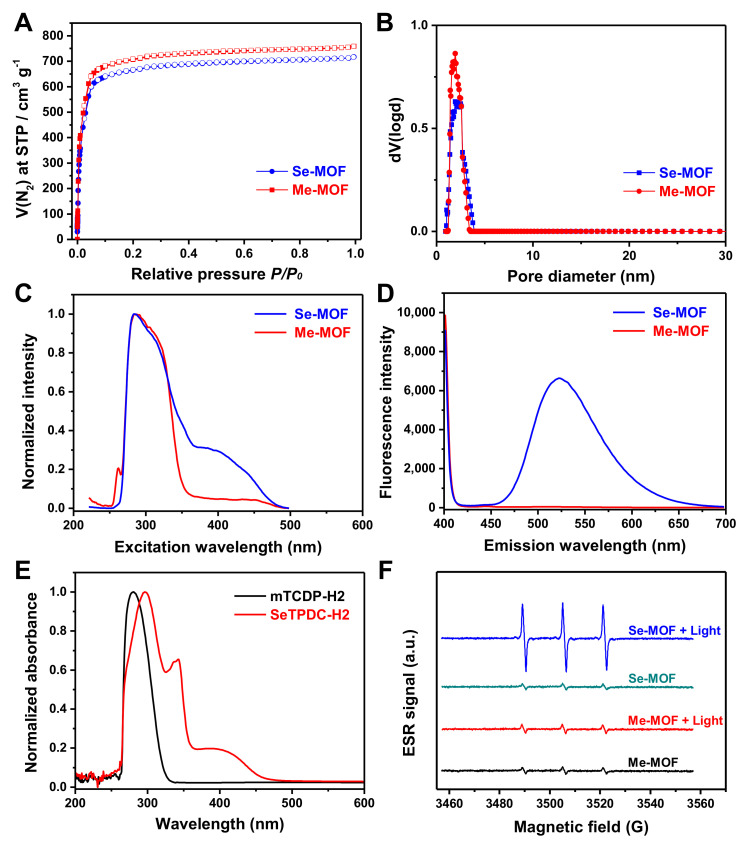
(**A**) Nitrogen sorption isotherms and (**B**) Barrett−Joyner−Halenda pore distribution of the Me-MOF and Se-MOF. (**C**) Excitation spectra (λ_em_ = 522 nm) and (**D**) emission spectra (λ_ex_ = 390 nm) of the Me-MOF and Se-MOF dispersed in DMF. (**E**) UV−vis absorption spectra of mTPDC-H2 and SeTPDC-H2 in DMF. (**F**) ESR spectra of a PBS solution (pH 7.4) containing Me-MOF or Se-MOF before and after light irradiation (450 nm, 3 mW cm^−2^).

**Figure 4 molecules-27-08908-f004:**
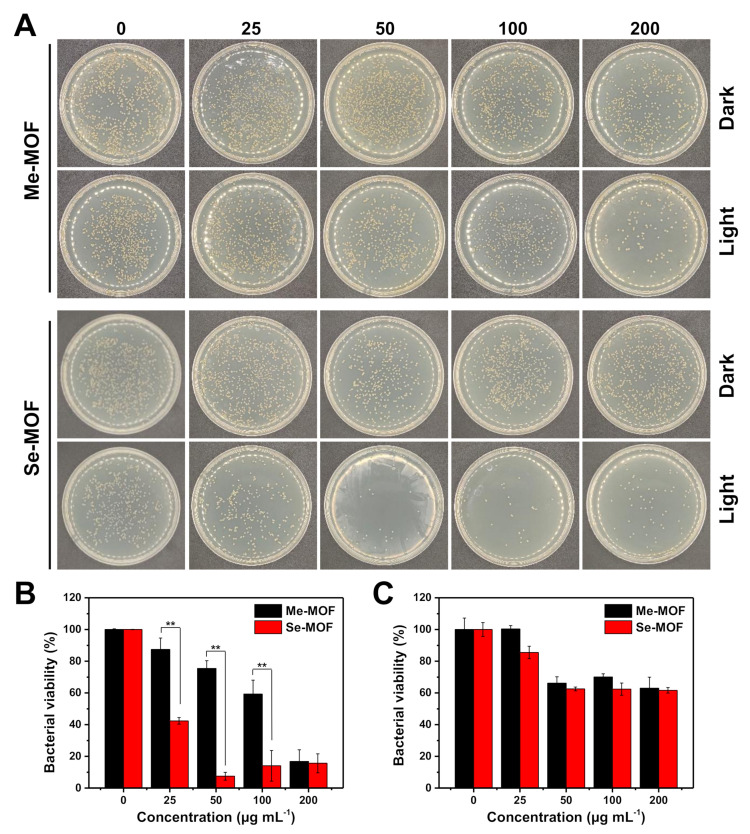
(**A**) Representative photographs of *S. aureus* colonies in LB agar plates after being incubated with different concentrations of Me-MOF or Se-MOF, followed by irradiation three times with a LED light (450 nm, 3 mW cm^−2^) for 20 min or incubation in the dark. (**B**,**C**) Percentage viability of bacteria *S. aureus* after different treatments, followed by light irradiation (**B**) or incubation in the dark (**C**). Data are presented as mean ± standard deviation (*n* = 3; ***p* < 0.01).

**Figure 5 molecules-27-08908-f005:**
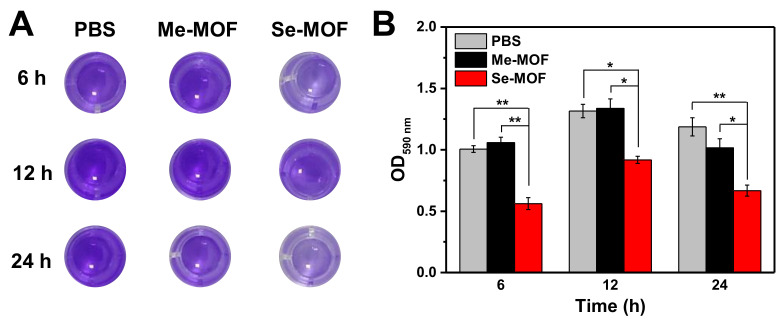
(**A**) Representative photographs of CV−stained biofilms after being incubated with Me-MOF or Se-MOF, followed by irradiation three times with LED light (450 nm, 3 mW cm^−2^) for 20 min. (**B**) Quantification of the OD measurement from the CV−stained biofilms in (**A**). Data are presented as mean ± standard deviation (*n* = 3; **p* < 0.05 and ***p* < 0.01).

## Data Availability

Not applicable.

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
