# Peer review of "Photodynamic Inactivation of Bacteria and Biofilms with Benzoselenadiazole-Doped Metal-Organic Frameworks"

_molecules, 2022, doi:10.3390/molecules27248908_

Round 1

Reviewer 1 Report

The authors described the manuscript entitled "Photodynamic Inactivation of Bacteria and Biofilms with Benzoselenadiazole-Doped Metal-Organic Frameworks" nicely. The authors discussed a strategy for the synthesis of zirconium (IV)-based benzoselenadiazole-doped MOFs with mixed ligands and demonstrated their effectiveness for the photo-dynamic inactivation of bacteria and biofilms. The use of MOFs is due to their unique properties, including tunable composition and size, large surface area, and high porosity for the delivery of various therapeutic agents.

The manuscript is well-written & organized but some points must be explained before publication:

1. Novelty and objective of the study should be clearly mentioned n the manuscript.

2.  The value of temperature should be written in Kelvin(in some places it is written as centigrade and in some other places in Kelvin). 

3. Why the photoactivity of Se-MOF was significantly higher than that of Me-MOF?

4.  Why Me-MOF has no inhibition effect on bacteria biofilms under 450-nm light irradiation but Se-MOF has significant inhibition under identical conditions? Explain.

5. Write the name of the manufacturer of chemicals from where you got them.

6. Mention the model & company name of the instruments used for the study.

7. Include C13 NMR spectra for better structural elucidation.

Reviewer 3 Report

1. The MOF structure are not characterized There is no NMR, Mass and Single crystal structure go the proposed structure. Without any characterization how ate authors are proposing the structure.

2. The author should provide the photo activity of the ligand Se TPDC-H2 against bacteria.  

3. The author should doe DPBF assay for both the ligands and both MOFs to quantify singlet oxygen quantum yield by comparing to the standards.

4. The author need to put reference to their statement "inaddition, incorporation of benzoselenadiazole-containing ligands into MOF skeletons can also avoid the aggregation of organic photosensitizers."

Round 2

Reviewer 1 Report

The authors have submitted the revised manuscript in light of the referee's comments. They answered all queries raised by the referee. Paper may be accepted for publication.

Author Response

Thanks for your recommendation of publication.

Reviewer 2 Report

Thank you for addressing all the comments. Overall, it is a nice work, and I suggest the manuscript be published in its current form.

Author Response

(The authors gave the same response as above.)

Reviewer 3 Report

Thanks to the authors for the updated version of the manuscript. It looks all the questions have been addressed except the PDT activity of the ligands. I believe the manuscript will make more sense if the could report the PDT activity of the ligands. I understand that the solubility of organic ligands will be poor, but the could use 0.5% DMSO to do the experiment. 

Author Response

Answer: As suggested, we have tried to investigate the PDT activity of the SeTPDC-H2 ligand in the culture medium containing DMSO (0.5%). However, when SeTPDC-H2 was added into a 0.5% DMSO solution, it precipitated immediately to form large aggregates. Based on the differences in physical and chemical properties between SeTPDC-H2 and Se-MOF, they may have different antimicrobial mechanisms via different approaches (including controlled release of antibacterial components, sizes, morphologies, and interaction with the bacterial cell wall). Thus, these investigations would not be able to provide information that either corroborates or refutes what we have already confirmed in this study. The related explanations and two references have been added in the revised manuscript (line 184-188).

In our study, we have demonstrated that the incorporation of benzoselenadiazole-containing ligands into MOF skeletons endowed them with high efficiency to produce 1O2 under 450-nm light irradiation, causing photodynamic inactivation of S. aureus bacteria and biofilms. This is a new strategy to construct the photosensitive MOFs with mixed ligands for photodynamic antibacterial applications. In addition, this strategy can also be applied to develop other MOF-based photodynamic agents, thus facilitating their transformation into clinical photodynamic antibacterial applications.